# Obstructive Sleep Apnea as a Risk Factor for COVID-19 Severity—The Gut Microbiome as a Common Player Mediating Systemic Inflammation via Gut Barrier Dysfunction

**DOI:** 10.3390/cells11091569

**Published:** 2022-05-06

**Authors:** Saif Mashaqi, Rekha Kallamadi, Abhishek Matta, Stuart F. Quan, Salma I. Patel, Daniel Combs, Lauren Estep, Joyce Lee-Iannotti, Charles Smith, Sairam Parthasarathy, David Gozal

**Affiliations:** 1Department of Pulmonary, Allergy, Critical Care and Sleep Medicine, The University of Arizona College of Medicine, Tucson, AZ 85719, USA; stuart_quan@hms.harvard.edu (S.F.Q.); salmapatel@arizona.edu (S.I.P.); combs89@arizona.edu (D.C.); lestep@arizona.edu (L.E.); spartha1@arizona.edu (S.P.); 2Department of Internal Medicine, The University of North Dakota School of Medicine, Grand Forks, ND 58203, USA; rekha.kallamadi@sanfordhealth.org (R.K.); abhishek.matta@sanfordhealth.org (A.M.); 3Division of Sleep and Circadian Disorders, Brigham and Women’s Hospital and Division of Sleep Medicine, Harvard Medical School, Boston, MA 02115, USA; 4Department of Sleep Medicine, The University of Arizona College of Medicine, Phoenix, AZ 85006, USA; joyce.lee-iannotti@bannerhealth.com; 5The Intermountain Healthcare, Merrill Gappmayer Family Medicine Center, Provo, UT 84604, USA; charles.smith2@imail.org; 6Department of Child Health, University of Missouri School of Medicine, Columbia, MO 65201, USA; gozald@health.missouri.edu

**Keywords:** COVID-19, obstructive sleep apnea, gut microbiome, zonulin, inflammation, tight junction, intestinal permeability, bacterial translocation, intermittent hypoxia

## Abstract

The novel corona virus that is now known as (SARS-CoV-2) has killed more than six million people worldwide. The disease presentation varies from mild respiratory symptoms to acute respiratory distress syndrome and ultimately death. Several risk factors have been shown to worsen the severity of COVID-19 outcomes (such as age, hypertension, diabetes mellitus, and obesity). Since many of these risk factors are known to be influenced by obstructive sleep apnea, this raises the possibility that OSA might be an independent risk factor for COVID-19 severity. A shift in the gut microbiota has been proposed to contribute to outcomes in both COVID-19 and OSA. To further evaluate the potential triangular interrelationships between these three elements, we conducted a thorough literature review attempting to elucidate these interactions. From this review, it is concluded that OSA may be a risk factor for worse COVID-19 clinical outcomes, and the shifts in gut microbiota associated with both COVID-19 and OSA may mediate processes leading to bacterial translocation via a defective gut barrier which can then foster systemic inflammation. Thus, targeting biomarkers of intestinal tight junction dysfunction in conjunction with restoring gut dysbiosis may provide novel avenues for both risk detection and adjuvant therapy.

## 1. Introduction

In December 2019, a novel coronavirus, now named (SARS-CoV-2), was described in Wuhan, Hubei Province, China as the cause of a pandemic disease that killed more than one million people in its first six months worldwide [1]. To date, nearly 500 million confirmed cases and more than six million patients have died as per the Johns Hopkins registry [2]. The virus affects predominantly the upper respiratory tract and lung tissues. The presentation of the disease is variable ranging from asymptomatic to the typical symptoms of fever, shortness of breath, and cough, nausea, diarrhea, headache, dizziness, sore throat, and generalized weakness, and in some to a severe illness characterized by acute respiratory distress syndrome (ARDS) and multi-organ dysfunction and ultimately death [3,4]. SARS-CoV-2 can cause a severe inflammatory reaction, initially secondary to the replication of the virus in the lungs, and intriguingly even after virus clearance (~day 14 post-infection), whereby the inflammatory response can even be more intense, suggesting activation and amplification of viral independent pathways [5].

Many risk factors have been identified and linked to the severity of COVID-19, among which age (>60 years), co-morbid diseases (such as obesity, hypertension, and diabetes mellitus), leukocytosis, lymphopenia, and computerized tomographic scan findings of the lungs are most prominent [6]. Obstructive sleep apnea (OSA) is a common sleep disorder, characterized by repetitive episodes of collapse in the pharyngeal airway, which is clearly associated with many co-morbid diseases, particularly cardiovascular, cerebrovascular, and metabolic conditions [7].

Recently, substantial interest has developed in the human microbiome (especially the gut microbiota) due to the emerging evidence linking the microbiome to chronic illnesses, such as cardiovascular, cerebrovascular, metabolic, malignant, and even psychiatric diseases. This interest is further exemplified by the human microbiome project that was initiated in 2007. The program received more than $170 million in funding to identify the characteristics of the human microbiota in different sites of the body and to determine the role of the microbiota in different diseases [8]. The term “microbiota” refers to the collection of all microbes (bacteria, viruses, fungi, and parasites) that populates in a specific environment or niche. Most of these microbes are found in the gut (~ 10^13^–10^14^ bacteria represented by 2000–3000 species) [9]. The five most common bacterial phyla that reside in the colon are Bacteroidetes, Firmicutes, Actinobacteria, Proteobacteria, and Cerrucomicrobia [10]. More than 90% of gut microbiota are Firmicutes (Gram-positive) and Bacteroidetes (Gram-negative). Normally, there is a healthy symbiotic relationship between the gut microbiota and the host. A perturbation in this balance secondary to external factors (such as diet, sedentary lifestyle, smoking, alcohol intake, and antibiotics) can shift the profile of microbial species towards more harmful species. This perturbation in the ecosystem between the microbiota and the host is called “dysbiosis”. The presence of dysbiosis can release proinflammatory and inflammatory mediators from these harmful microbial species fostering inflammation and predisposition for co-morbid illness. The classical example is lipo-polysaccharides (LPS) produced by Gram-negative bacteria that can escape a leaky gut barrier to the systemic circulation, mounting a state of systemic inflammation which can be the cornerstone for many chronic diseases [11,12].

In this critical review of the literature, we explore the hypothesis that the gut microbiota mediates worse clinical outcome in patients with COVID-19 infection who have untreated OSA. This will shed light on the gut microbiota as a potential novel adjuvant interventional target in patients with OSA who are infected with COVID-19. We will initially examine evidence associating COVID-19 and OSA and whether OSA is a risk for worse COVID-19 outcome. Then we will review the role of the gut microbiota and dysbiosis in both diseases, first in clinical studies and then in experiments conducted at the cellular level looking for markers of tight junction dysfunction and microbial translocation across the gut barrier. Finally, we will explore the associations among COVID-19, OSA and gut dysbiosis that may explain severity of COVID-19 illness.

## 2. OSA and COVID-19 (Is There a Plausible Link Clinically?)

Several co-morbid conditions are risk factors for increased severity of COVID-19 infection as well as great morbidity and mortality. A meta-analysis published recently which included 18 studies concluded that hypertension, type-II diabetes, cardiovascular diseases, chronic obstructive pulmonary disease, chronic kidney disease and cancer have a higher risk of more severe COVID-19 infection compared to those without co-morbid diseases [13]. It is well known that OSA has been implicated as a causal factor for many of these co-morbid diseases (especially hypertension, diabetes, and cardiovascular diseases) and treatment with positive airway pressure can mitigate some of these chronic diseases [14,15,16,17,18].

Cade and Gottlieb et al. [19,20] were one of the earliest groups that investigated OSA as a risk factor for worsening COVID-19 outcome (intubation, intensive care unit admission and overall mortality) in hospitalized patients. After adjustment for demographic factors, BMI, and medical co-morbidities; the only significant outcome that remained was overall mortality. Maas et al. [21] had similar findings and concluded that OSA was associated with increased risk of hospitalization and developing respiratory failure. Furthermore, they noticed an eight-fold risk of COVID-19 infection in patients with OSA compared to a similar age population. On the other hand, Strausz et al. [22] could not confirm these findings in their large biobank database (FinnGen study). They concluded that OSA was associated with higher risk of hospitalization in patients with COVID-19 but did not notice an increased risk of contracting COVID-19 infection in these patients. However, their sample size was limited, and this was likely a contributing factor. Kravitz et al. [23] concluded that patients with moderate or severe OSA had a higher rate of COVID-19 hospitalization compared to those with mild sleep apnea. However, the association was weakened when adjusting for confounders. Beltramo et al. [24] compared clinical outcomes in patients with OSA who were hospitalized with COVID-19 with influenza infection. They noticed that patients with OSA and COVID-19 had higher rates of pulmonary embolism, ventilator-associated pneumonia, Intensive Care Unit (ICU) stay, and death compared to patients hospitalized for influenza alone. OSA is a common sleep disorder in patients with interstitial lung diseases [25]. It is interesting to study the impact of OSA on COVID-19 outcome in these patients. Among a South Korean population, Tak Kyu Oh et al. [26] investigated the effects of chronic respiratory disease on the risk of COVID-19 and mortality. COPD, asthma, interstitial lung disease, lung cancer, OSA, and tuberculosis were considered as chronic respiratory diseases. They concluded that patients with OSA and interstitial lung diseases had a higher incidence of COVID-19 infection, while OSA alone did not show a higher mortality rate in hospitalized COVID-19 patients. Cariou et al. [27] conducted one of the largest retrospective studies that was intended primarily to investigate the impact of type II DM on COVID-19 clinical outcome (CORONADO study). The primary outcome was combined tracheal intubation for mechanical ventilation and/or death within seven days of admission. Surprisingly, treated OSA was an independent risk of death at day seven after admission in COVID-19 patients. Recently, in a study of all community-dwelling Icelandic citizens Rögnvaldsson et al. [28] found a two-fold increased risk of severe COVID-19 in those with OSA with slight attenuation after adjustment for demographic characteristics and various comorbidities. Some studies used prediction models to assess the risk of OSA in patients infected with COVID-19 and found that patients with high OSA risk tend to have higher odds of in-hospital death and ICU transfer rate [29].

In contradistinction, some studies have not identified OSA as an independent risk factor for worse COVID-19 outcomes. Goldstein et al. [30] reported that the prevalence of sleep disordered breathing (SDB) in 572 patients hospitalized for COVID-19 was close to 20%. However, after adjusting for age, sex, body mass index, and race, no significant relationship was apparent between OSA and mechanical ventilation, treatment with vasopressors, length of stay, and death. Mashaqi et al. [31] had similar results from a retrospective review of 1738 patients hospitalized with COVID-19 during an 8-month period in 2020. They reported that OSA did not appear to be an independent risk factor for worse COVID-19 outcomes in hospitalized patients after adjustment for demographic factors, BMI, and co-morbidities (ICU admission, intubation and mechanical ventilation, mortality, and length of hospital-stay). More recently, Pena Orbea et al. [32] conducted a retrospective study in a large Cleveland, OH area healthcare system to investigate the impact of SDB on COVID-19 positivity and COVID-19 clinical outcomes This study showed that only sleep-related hypoxia ((otal sleep time < 90) but not SDB ((AHI > 5 events per hour)) was associated with worse COVID-19 related outcomes. Similarly, other studies did not show OSA to be an independent risk factor for COVID-19 severity [33,34,35]. The variation in results among these studies can be attributed to several factors such as the retrospective nature of these studies (selection bias, confounding variables, etc.), the lack of objective evidence (i.e., polysomnography [PSG] or home sleep apnea test [HSAT]) in diagnosing OSA and reliance on data from self-reports or electronic medical records, and finally small sample size in many studies.

All the above were retrospective cohorts or case control studies. Very few prospective studies have been conducted, but nevertheless similarly concluded that OSA was associated with clinical worsening of COVID-19, increased O2 requirement, increased mortality, and poor outcome [36,37]. Table 1 summarizes these clinical studies.

We conclude that many of the aforementioned studies have shown that OSA is a risk factor for worse COVID-19 clinical outcome, even after adjustment for several variables. However, more prospective trials are required to confirm this observation. Treating OSA with positive airway pressure therapy and its impact on COVID-19 has not been studied and needs to be thoroughly evaluated.

## 3. OSA and Gut Dysbiosis (Is There Evidence?)

The main features in the pathophysiology of OSA consist of intermittent hypoxia (IH) and hypercapnia and sleep fragmentation. IH is reflected by oscillation in the partial oxygen pressure (PaO_2_) in the blood vessels and the capillaries, especially the capillaries of the gut which potentially alters the gut microbiota profile [38]. Most experiments conducted in this field utilized animal models and employed a hypoxia chamber to mimic an OSA environment. Animals are exposed to these oscillatory PaO_2_ changes in the hypoxia chamber for a specific period of time. This creates a PaO_2_ gradient between the surface close to the gut epithelium and the center of the intestinal lumen which foster changes in the gut microbiota by expressing more facultative and obligate anaerobic bacteria [38].

Morenos-Indias et al. [39] conducted an important experiment in this field. Mice were exposed to air with cyclic changes in partial oxygen pressure (normal air at 21% oxygen for 40 s followed by low PO_2_ air at 5% oxygen for 20 s). After six weeks, fecal samples from IH-exposed mice and corresponding controls were analyzed using 16S r RNA pyrosequencing. An increase in the relative abundance of Firmicutes, Prevotella, Paraprevotella, Desulfovibrio, and Lachnospiraceae and reduced Bacteroidetes, Odoribacter, Turicibacter, Peptococcaceae and Erysipelotrichaceae were observed in mice with IH compared to controls respectively. Importantly, reversing IH by exposing these mice to normoxic conditions for six weeks did not restore the gut dysbiosis [40]. Some of the bacteria that increased their populations after IH have been linked to metabolic, inflammatory, and even neoplastic diseases. For instance, Prevotella is a Gram-negative bacterium, that produces LPS which fosters inflammation [12]. Desulfovibrio produces hydrogen sulfide which has been shown in some studies to increase the risk of colon cancer [41]. Conversely, Bacteroidetes are a very important source for short-chain fatty acids (butyrate, acetate, and propionate) that act as a nutrient source for colonic epithelia cells [42]. Wang et al. [43] had similar results of 10 weeks of IH on the regulation of *Akkermansia mucinphila*, *Clostridium spp*., *Lactococcus spp*. and *Bifidobacterium spp*. and their metabolites, such as tryptophan, free fatty acids, branched amino acids, and bile acids which might play a synergistic role with high-fat diet on hyperlipidemia. Tripathi et al. [44] had similar results and concluded that IH and hypercapnia increased the relative abundance of Mogibacteriaceae, Oscillospira, Lachnospiraceae and Clostridiaceae. These species can alter primary and secondary bile acids and decrease levels of unsaturated fatty acids (such as elaidic acid) which can increase the risk of cardiovascular diseases and atherosclerosis [45]. Liu et al. [46] showed the role of IH in synergism with high-salt diet in worsening hypertension. They noticed a decrease in the relative abundance of *Lactobacillus rhamnosu* which can elevate blood pressure via an increase in trimethylamine *N*-oxide (TMAO) and inflammatory mediators such as IFN -γ. Another OSA animal model used implantation of an endotracheal obstruction device that can inflate and deflate mimicking the repetitive episodes of upper airway obstruction in OSA and its consequent oscillations in PaO_2_. Durgan et al. [47] concluded that a combination of IH and high-fat diet (HFD) induced an increase in blood pressure readings in rats that was statistically significant compared to control or IH alone or HFD alone. The microbiota profile in fecal samples of the IH+HFD rodents were characterized by an increase in the Firmicutes/Bacteroidetes ratio and a decrease in the *Ruminococcaceae* at the family level and the *Clostridiales* at the order level. Table 2 provides a summary of these animal experiments.

In humans, the scarcity of clinical trials conducted in this field owes to the extremely challenging nature of controlling for many variables that affect the gut microbiome. Collado et al. [50] conducted a clinical trial in two-year-old snoring children and observed a shift in the gut microbiota with a higher Firmicutes/Bacteroidetes ratio and a lower Actinobacteria/Proteobacteria ratio. Recently, Wang et al. [51] examined the gut microbiota in 32 patients diagnosed with OSA using PSG and concluded that OSA patients have a disproportionate Firmicutes/Bacteroidetes ratio with increased Firmicutes and decreased Bacteroidetes in the gut microbiota compared to the healthy population. Furthermore, at the family level they noticed an increase in the abundance of *Rikenellaceae* and *Alistipes* and decrease in the *Clostridium_XlVa* in OSA patients compared to control. They also concluded there was an imbalance in the proportion of T helper type 17 (Th17) and T regulatory (Treg) cells in blood using flowcytometry in OSA group which might be related to the shift in gut microbiota.

The aforementioned studies clearly support an association between OSA (specifically IH) and gut dysbiosis. However, it should be emphasized that causality has not been established and more studies are warranted to prove this concept.

## 4. COVID-19 and the Gut Microbiome

The relation between the shift in the gut microbiota balance and SARS-CoV-2 can be described as bidirectional. It is known that the immune system can be altered through the gut–brain axis which represents a two-way interaction between emotional and cognitive regions of the brain and intestinal function. Gut dysbiosis can therefore worsen the infection outcome. SARS-CoV-2 can directly change the microbiota profile (e.g., bacteria, fungi, or viruses). This includes the microbiome in different niches of the body (such as oral, lung, skin, upper air way, etc.). However, as mentioned earlier, since most of the microbiota are in the gut, we will focus in this section on the impact of SARS-CoV-2 on the gut microbiota.

Zuo et al. [52] compared the gut microbiota from fecal material in patients with community-acquired pneumonia and COVID-19 pneumonia and observed a significant depletion of beneficial commensals and enrichment of opportunistic pathogens (e.g., *C. hathewayi, B. nordii, A. viscosus*). These bacteria are associated with more severe COVID-19 disease since they highly predispose the host to secondary bacterial infection. Interestingly, *A. viscosus* is an upper respiratory tract opportunistic pathogen suggesting a connection between extra-intestinal sites and the gut. Furthermore, they noticed that four species from the genus *Bacteroides* of the phylum Bacteroidetes (*B. dorei**, B. thetaiotaomicron, B. massiliensis, and B. ovatus*) showed inverse correlation with the fecal viral load of SARS-CoV-2. *B. dorei* is known to suppress the expression of ACE2 on the surface of colonocytes. It is well known that ACE2 modulation on the surface of colonocytes facilitates SARS-CoV-2 entry into a colonocyte [53].

Similar to the gut bacteria, gut fungi can be disrupted secondary to COVID-19 infection. Zuo et al. [54] showed a heterogeneity in the gut mycobiome in the stool samples of COVID-19 +ve patients compared to COVID-19 −ve. In particular, *Candida* (*C. albicans*) and *Aspergillus* (*A. flavus* and *A. niger*) were enriched in the stool of these patients. *C. albicans* is usually associated with severe inflammatory reactions (whether gastro-intestinal or extra gastro-intestinal). *A. flavus* and *A. niger* are also associated with severe disease and most patients in the study who had these opportunistic pathogens in their stool were admitted to the ICU. Emphasizing the connection between extra-intestinal sites and the gut, patients who had *A. flavus* (respiratory pathogen) in the stool had cough before hospitalization. Their final conclusion was that COVID-19 pneumonia can cause alterations in the gut mycobiome, although these alterations are less dysbiotic (i.e., less change in diversity and richness) in comparison to alterations in the gut mycobiome induced by community-acquired pneumonia.

The same group studied the gut virome (DNA and RNA) in patients infected with COVID-19 with a broad severity spectrum (asymptomatic, mild, moderate, and severe) and compared them to non-COVID-19 controls. Although they did not notice a correlation between fecal SARS-CoV-2 levels and COVID-19 severity, they found underrepresentation of Pepper mild mottle virus (the most common plant-derived RNA virus that infects humans) and multiple bacteriophage lineages (DNA viruses). Conversely, they noticed enrichment of environment-derived eukaryotic DNA viruses and specific bacteriophages (Escherichia virus phage and Enterobacter phage). This suggests that the gut virome can play an important role in shaping host physiology and immune response against SARS-CoV-2 infection. Such alterations persisted even after discharge from the hospital (up to 30 days from disease resolution) which may suggest a role of the gut virome in patients with post-COVID-19 syndrome. More interestingly, they also noticed a correlation between some DNA-species and age. (Myxococcus phage, Bacteroides phage, Murmansk poxvirus, and Sphaerotilus phage) inversely correlated with human age which may explain the worse outcomes of COVID-19 in elderly patients [55].

Yeoh et al. [56] examined the disturbance in gut microbiota secondary to COVID-19 infection and its relationship to the severity of inflammation and inflammatory markers. They noticed a depletion in some bacterial taxa that can play an important role in reducing inflammatory aggressiveness (*B. adolescentis, F. prausnitzii, E. rectale, R. (Blautia) obeum and D. formicigenerans)* and an increase in the concentrations of *TNF-α, CXCL10, CCL2 and IL-10.* These finding are in agreement with previous studies suggesting a role for the gut microbiota in modulating the host immune response.

Importantly, the impact of COVID-19 on the gut microbiome (commensals or opportunistic pathogens) persisted even after the nasopharyngeal clearance of the SARS-CoV-2 post-hospitalization. (An example of this phenomenon was the continued presence of *Aspergillus* lineage) in feces [54]. This suggests that the persistent alteration in the gut microbiome post-recovery from COVID-19 infection may explain some of the immune-related symptoms (e.g., fatigue, joint pain, etc.) that can persist up to six months in some patients.

Mazzarelli et al. [57] concurred with Zuo et al. [52] and confirmed the variations in relative abundance and diversity of the gut microbiota in COVID-19 patients with different severity. They concluded that most of the changes (the lowest microbial richness) were noted in ICU patients (most severe patients). There was an increase in *Erysipelotrichaceae* (which is associated with inflammatory diseases affecting the gastrointestinal (GI) system) [58] in COVID-19 patients in ICU and reduction in *Faecalibacterium* (which is associated with Crohn’s disease and colorectal cancer) [59], and *Ruminococcaceae, Clostridiaceae* (involved in short-chain fatty acids (SCFAs) production especially butyrate) [60].

Other opportunistic pathogens that are shown to predominate in the gut microbiota of COVID-19 patients include (*Streptococcus, Rothia, Veillonella, Erysipelatoclostridium, and Actinomyces*) [61]. These taxa are associated with higher levels of inflammatory markers (CRP, IL-6, and TNF-α). For example, *Rothia* was reported in some studies to be associated with pneumonia in immunocompromised patients and with secondary bacterial infection in patients with avian H7N9 influenza infection [62,63]. Actinomyces may aggravate inflammation caused by inflammatory bowel disease [64].

Zuo et al. [65] concluded that SARS-CoV-2 after being cleared from the respiratory tract, can remain active in the GI tract even before the appearance of GI symptoms suggesting that the life cycle of SARS-CoV-2 in the GI tract is longer than in the respiratory tract. This lagging period between the clearance of the virus from the respiratory tract and the GI symptoms can be up to one week or even longer. Interestingly, they concluded that COVID-19 patients with fecal samples containing high SARS-CoV-2 infectivity (i.e., higher 3ʹ vs. 5ʹend coverage) have a greater abundance of microbiota with high functional capacity for nucleotide de novo biosynthesis, amino acid biosynthesis and glycolysis (*Collinsella aerofaciens, Collinsella tanakaei, Streptococcus infantis, Morganella morganii*). For example, adenosine and guanosine are two important metabolites involved in purine metabolism. Adenosine levels can increase during inflammation and hypoxia and guanosine can induce cytokine production [66]. Some amino acids (e.g., *L*-serine) can expand the inflammatory reaction in some inflammatory diseases [67]. Conversely, fecal samples that contain low-to-none SARS-CoV-2 infectivity have higher abundance of SCFA producing bacteria (*Parabacteroides merdae, Bacteroides stercoris, Alistipes onderdonkii, and Lachnospiraceae bacterium 1_1_57FAA*). A summary of these studies is located in Table 3.

## 5. OSA and SARS-CoV-2, Similar Pathological Features on the Gut Barrier at the Cellular and Molecular Level

A common feature of both OSA and SARS-CoV-2 infections is inflammation. Although the severity of inflammation in COVID-19 is usually more intense compared to OSA, a common target in both diseases is the gut barrier leading to what is known as “leaky gut syndrome”. Before reviewing the literature in this regard, we would like to briefly describe the anatomical, histological, and cellular structures of intestinal epithelial cells. Then we will review different biomarkers of gut barrier dysfunction that were studied in COVID-19 and OSA.

The small and large intestines are composed histologically of a single layer of simple epithelium that is responsible for absorbing water and nutrients while simultaneously acting as a barrier against pathogens. In the small intestine, the epithelium is differentiated into epithelial protrusions called villi that form the majority of the epithelial surface area where the absorption process takes place. The main cell types that cover these villi are absorptive enterocytes, mucous-secreting goblet cells, and hormone-secreting enteroendocrine cells. Crypts which are found at the bottom of the villi contain Paneth cells which play a critical role in innate immunity and antibacterial defense. In the large intestine, the epithelium is composed only of crypts with no villi and does not contain Paneth cells [68]. The intestinal epithelial cells are connected inter-cellularly by tight semi-permeable gates that allow the influx of ions and prevent the passage of pathogens. As shown in Figure 1, these adjacent structures are called “apical junctional complexes” (AJC). AJCs are composed of tight junctions, and adherens junctions [69]. Tight junctions are composed of trans-membrane proteins that act as a fence and a gate [70]. The gate function regulates the influx of ions and solutes that are less than 600 Da paracellularly (between cells) [71], and the fence function prevents the mixing of membrane proteins between the apical and basolateral parts of the cell (i.e., preserves the cell polarity). These functions are maintained by the trans-membrane protein (claudin). Another trans-membrane protein of the tight junctions called (occludin) regulates the paracellular permeability barrier between cells. The other group of AJCs are adherens junctions which include several trans-membrane proteins and glycoproteins that initiate and maintain cell–cell adhesion and regulate actin cytoskeleton. These include E-cadherin and catenins [70]. Tight junctions and adherens junctions are anchored to cytoplasmic proteins via a group of connector proteins called ZO proteins. ZO-1 is an example of this group which can link between tight junctions and adherens junctions in addition to anchoring other proteins in the ZO protein group together such as ZO-2 and ZO-3 [70].

One of the early and first identified proteins released by enterocytes which play an important role in the regulation of intestinal permeability is zonulin. Zonulin is a 47-kDa protein that has been shown to increase intestinal epithelial permeability in non-human experiments [72]. Zonulin is the only precursor for haptoglobin-2 which regulates tight junction permeability. This occurs via epidermal growth factor receptor (EGFR) transactivation through proteinase-activated receptor 2 (PAR2) [73]. Zonulin is overexpressed in some autoimmune diseases (such as celiac disease and type-I diabetes mellitus) [74,75,76]. The main stimuli for zonulin release are bacteria and gliadin, a protein component of gluten. Historically, *Vibrio cholerae* and the discovery of zonula occludens toxin (Zot), an enterotoxin elaborated by this bacterium, helped further understand the role of bacteria in regulating intestinal permeability [77]. As shown in Figure 2, zonulin transactivation of EGFR via PAR2 activates phospholipase C, which in turn, activates a cascade of reactions that lead to an increase in the intracellular Ca and activation of protein kinase C alpha (PKCα). Activated (PKCα) enhances the phosphorylation of ZO-1, ZO-2, and myosin 1C in addition to the polymerization of F-actin. As a result, ZO-1 is displaced from the junctional complex which leads to a loosened junction and increased permeability [78].

Prasad et al. [79] (published as preprint manuscript) examined the plasma microbiome in 30 COVID-19 patients (mild, moderate, and severe cases) in comparison to healthy controls. At the phylum level, Proteobacteria was highest in all samples (22%–91%), followed by Firmicutes (10%–71%), and Actinobacteria (6%–27%). Bacteroidetes was present in a very low percentage. At the genus level, Gram-negative bacteria (Acinetobacter, Nitrospirillum, Cupriavidus, Pseudomonas) were higher than Gram-positive bacteria (Staphylococcus and Lactobacillus). Several gut permeability markers were tested. Fatty acid-binding protein-2 (FABP2) levels were higher in the plasma of COVID-19 patients. FABP2 is an intra-epithelia protein that binds free fatty acids and is involved in lipid transport. During intestinal epithelial cell damage, this protein is released into the circulation and its level is increased. Other gut permeability markers that were measured are peptidoglycan (PGN) and lipopolysaccharides (LPS) and both were elevated in the plasma of COVID-19 patients compared to healthy controls. Studies that utilized plasma microbiome are limited [80,81]. Most studies used plasma metabolome and fecal microbiome. Since the fecal viral shedding continues for more than 30 days in some COVID-19 cases after lung recovery [82], the blood microbiome findings in the context of the gut permeability markers suggest translocation of gut bacteria from the colon into the circulation.

Similarly, Giron et al. [83] conducted another analysis in 60 patients who tested positive for SARS-CoV-2 (mild, moderate, and severe) and 20 healthy controls. The authors found that patients with severe COVID-19 had high levels of markers of tight junction permeability and microbial translocation compared to patients with mild COVID-19 and healthy controls. These included zonulin and occludin. Subsequently, microbial products (lipopolysaccharide binding protein (LBP), β-glucan (polysaccharide cell wall component of most fungal species) were higher in patients with severe COVID-19 infection. Authors also measured inflammatory markers and immune modulation markers. They confirmed an increase in monocyte inflammation marker (sCD14), neutrophil inflammation marker (MPO), and soluble CD163 (sCD163) in patients with severe COVID-19. Furthermore, they noticed an increase in the levels of many cytokines and chemokines (such as IL-6, IL-1b, MCP-1, IP-10, and TNF-a), markers of inflammation and thrombogenesis (such as C-reactive protein (CRP) and D-dimer), a marker of complement activation (C3a), a marker of oxidative stress (GDF-15), and three immunomodulatory galectins (galectin-1, 3, and 9). Importantly, there was a strong positive correlation between the tight junction permeability markers levels (zonulin, LBP, or β-glucan) and the severity of systemic inflammation and immune activity. To further confirm the findings of tight junction permeability markers levels in patients with severe COVID-19 infection, the authors conducted an analysis of an independent validation cohort. A combination of zonulin, LBP, and sCD14 was able to distinguish hospitalized from non-hospitalized individuals in the validation cohort with an area underneath the curve of 88.6% (95% confidence interval 0.80–0.96).

Yonker et al. [84] concurred with the above results and conducted a study in 19 children who developed multisystem inflammatory syndrome (MIS-C) which is a severe, life threatening complication of SARS-CoV-2 infection and compared them to 26 children with acute COVID-19 and 35 healthy controls. Most children with MIS-C developed the symptoms after the resolution of respiratory symptoms and their nasopharyngeal swabs were negative for SARS-CoV-2. Almost 90% of these children presented with GI symptoms (such as nausea, diarrhea, vomiting, and abdominal pain) compared to only 27% of children with acute COVID-19 presenting with GI symptoms. Most of the children with MIS-C had detectable viral loads in their stool samples suggesting that there was another nidus of infection contributing to their severe and life-threatening symptoms. The authors noticed an increase in zonulin, LBP, and sCD14 levels in children with MIS-C while there were normal levels in children with acute COVID-19 infection and healthy controls. This provides evidence of gut barrier breakdown and tight junction dysfunction in severe COVID-19 infection that leads to microbial translocation and systemic inflammation. Authors in this study were able to detect SARS-CoV-2 viral components in the circulation (including spike, S1, and nucleocapsid antigens) several weeks after initial infection.

Guo et al. [85] created an intestinal model on a biochip to simulate the pathophysiological changes associated with SARS-CoV-2 infection. They noticed that SARS-CoV-2 caused a) severe destruction and disruption in adherent junctions, identified by E-cadherin expression and VE-cadherin immunostaining, b) damage to the intestinal villus-like structures, c) disruption in the intestinal mucus layer, and finally d) decreased density and size of endothelial cells. The injury to the endothelial cells and the adherent junctions between them can explain some of the thrombogenic and vascular complications seen in patients with COVID-19 [86]. However, they concluded that the injury of SARS-CoV-2 to the epithelial cells is greater than the endothelial cells possibly due to higher viral loads. They also confirmed an extensive increase in cytokines and chemokines in both epithelial and endothelial cells (TNF, IL-6, CXCL10, CCL5, and CSF3).

In vitro experiments confirmed the influence of IH (hypoxia/reoxygenation) and sleep fragmentation on gut permeability and tight junction dysfunction. This impact was measured using transepithelial electrical resistance (TEER) or FITC-dextran [48,49,87].

In human studies, Kheirandish-Gozal et al. [88] concluded that LBP levels are elevated in children with OSA suggesting that markers of systemic inflammation are elevated in OSA. The authors also found that obesity played a synergistic role to the effects of OSA. Li et al. [89] measured plasma *D*- lactic acid (*D*-LA) and intestinal fatty acid-binding protein (I-FABP) levels using colorimetry and ELISA, respectively as biomarkers of gut barrier integrity in adults with OSA (defined as AHI ≥ 5 events per hour) and those without OSA (AHI < 5 events per hour). They confirmed dramatically higher levels of plasma (D-LA) and (I-FABP) in OSA patients compared to controls. Barcelo et al. [90] had similar results with (I-FABP) levels in patients with OSA. Furthermore, they measured circulating levels of zonulin and although zonulin levels were similar between groups, zonulin levels correlated negatively with the mean nocturnal oxygenation saturation (*p* < 0.05).

These studies suggest that both COVID-19 and OSA share the same features of in-creasing biomarkers of tight junction dysfunction and intestinal permeability.

## 6. Discussion and Future Directions (Connecting the Dots)

The putative association between OSA and COVID-19 severity is based on the similarities in the pathophysiology between these entities. The severe inflammatory response to COVID-19 is secondary to the replication of the virus in type-II pneumocytes and recruitment of virus independent pathways (e.g., ACE-2) [91]. Obstructive sleep apnea. which is characterized by IH and SF, has been studied over the last several decades using animal and human models. A major observation is the association between IH and low-grade systemic inflammation. This is a multifactorial process that includes several steps involving oxidative stress secondary to the excessive generation and propagation of reactive oxygen species, and induction of transcriptional pathways underlying many pro-inflammatory mediators and inflammatory cytokines (e.g., IL-6, TNF-α, CRP, IL-10) ultimately resulting in systemic inflammation [92]. However, similar to COVID-19, OSA displays substantial heterogeneity in the magnitude of the inflammatory response. One of the key factors in triggering inflammation is gut dysbiosis and in this review, we shed light on this critical concept in both diseases, OSA and COVID-19, and suggest that the gut microbiome acts as a mediator for worse clinical outcome in patients with COVID-19 who have a diagnosis of OSA (Figure 3).

The gut microbiota plays an important role in signaling the immune system and regulating the inflammatory process. In animal models, intact microbiota enhances the recognition of different pathogens by the immune system via specific receptors (e.g., toll-like receptors and nucleotide-binding oligomerization domain-like receptors) expressed on the surface of immune cells. For example, macrophages isolated from microbiota-depleted mice released lower levels of TNFα and higher levels of IL-10 in response to lipopolysaccharide [93]. Splenic dendritic cells isolated from germ-free mice and stimulated with LPS expressed less IL-5, IL-6 and TNFα [94]. Similarly, NK cells isolated from germ-free mice displayed compromised immune reactivity [95]. T and B cell numbers were lower in germ-free animal models with lower IgA, and IgG secretion. Moreover, T-cells (CD4 and CD8) showed a shift towards more CD4^+^ (TH-2) helper cells. Furthermore, gut microbiota metabolites, such as butyrate, propionate, and acetate, play an essential role in immunity and systemic inflammation. For example, butyrate is a key source of nutrients to the colonocytes [42,96,97].

Over the last few decades, the concept that intestinal permeability and gut barrier dysfunction are important in the regulation of the inflammatory reaction to a specific stimulus has been generally accepted. In this review, we provided evidence that OSA (in animal experiments) and COVID-19 are clearly associated with tight junction dysfunction and microbial translocation into systemic circulation thus propagating systemic inflammation. Since gut barrier dysfunction can be mediated through the gut microbiome, this offers additional evidence to the plausible link between the gut microbiome (as a mediator) and COVID-19 patients with untreated OSA. An important question in this regard is, what is the implication of these observations and how can they be helpful to patients clinically?

Conceptually, treating OSA would be the ideal therapeutic path towards reducing any adverse synergistic effect produced by the presence of OSA and COVID-19. However, currently it is unclear whether OSA is an independent risk factor for severity of COVID-19 infection. Furthermore, continuous positive airway pressure (CPAP) which is the gold standard for treating OSA is challenging in many patients and adherence to CPAP therapy is extremely variable [98]. Therefore, alternative OSA therapies such as dental devices [99] and medication regimens acting on the upper airway might help in reducing the severity of COVID-19 in patients with OSA who cannot tolerate PAP therapy [100].

Targeting the gut microbiome is another potentially novel approach. Most of the therapeutic interventions in this field are still experimental and their clinical applications are limited yet promising. The potential link between changes in gut microbiome with advancing age and the poorer outcomes associated with aging has also been noted [101,102], leading to the proposal of interventions based on probiotics as a vehicle to reduce risk [103]. Interestingly, most if not all of the risk factors identified as adversely affecting COVID-19 outcomes (hypertension, diabetes mellitus, obesity, and aging) are linked to gut dysbiosis. Gut dysbiosis (mainly an increase in Firmicutes/Bacteroides ratio) and particularly when combined with a high-fat diet will increase blood pressure in both animal models and in humans [47,104]. Furthermore, fecal matter transfer (FMT) from hypertensive mice into germ-free mice resulted in increased blood pressure values in recipient mice, suggesting a causal relationship between gut dysbiosis and hypertension [47]. Similarly, FMT of fecal material from IH-exposed mice to naïve mice results in sleep perturbations and excessive daytime sleepiness in the recipient mice as well as in metabolic dysfunction [105,106]. The latter findings concur with other experiments showing that gut dysbiosis can alter energy harvesting, fat deposition, insulin sensitivity, and modify systemic inflammatory measures and lead to increased risk of obesity and diabetes mellitus [107]. Although our understanding of the association between aging and the gut microbiome is still evolving, dietary changes, decreased physical activity, and other age-related physical changes affecting the overall gastrointestinal system also influence and modify the gut microbiota [108].

Another promising therapeutic avenue is targeting intestinal permeability and microbial translocation markers. Zonulin antagonist AT1001 (larazotide acetate) is currently in phase III trials for treatment of refractory celiac disease [109]. Larazotide acetate improved intestinal epithelial function in vivo and in vitro [110,111]. However, the role of zonulin antagonist AT1001 (larazotide acetate) and other targets for intestinal permeability markers in lessening COVID-19 severity still remain to be elucidated. Interestingly, Yonker et al. [84] obtained US FDA approval of larazotide acetate to treat a refractory sick 17-month-old boy with multisystem inflammatory syndrome (MIS-C) after he failed all anti-inflammatory therapies. The patient improved clinically (fever declined), inflammatory mediators and cytokines subsided, and spike antigen, nucleocapsid protein levels dropped. This case report suggests that improving intestinal permeability in some COVID-19 cases may be a useful adjuvant to more conventional treatment.

## 7. Conclusions

Based on the similarities between COVID-19 risk factors and OSA-associated comorbid diseases and the common background of inflammation and immune modulation in COVID-19 and OSA, gut dysbiosis should strongly be considered as a mediator of systemic inflammation and immune system modulation in COVID-19 in the presence of OSA. We postulate that OSA-induced gut dysbiosis may constitute a risk factor determining the severity of COVID-19 infection. Preliminary evidence suggests that markers of tight junction dysfunction might serve both as biomarkers of disease activity and as a target for intervention in conjunction with restoration of the gut microbiota.

## Figures and Tables

**Figure 1 cells-11-01569-f001:**
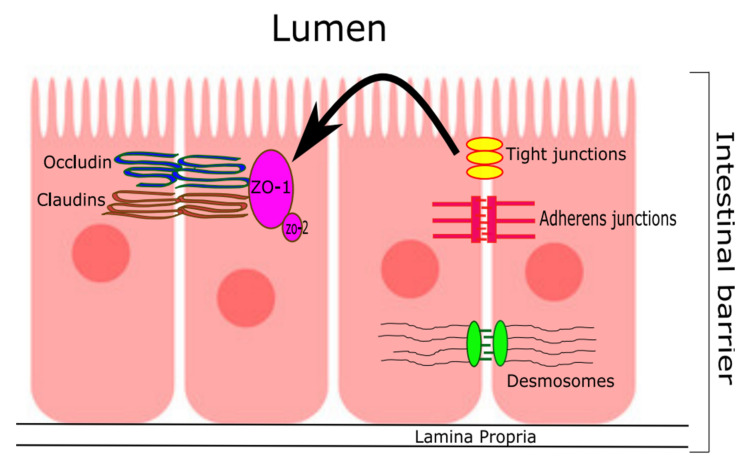
The apical junctional complex and tight junctions in the small intestine. ZO-1 = Zonula occludens-1; ZO-2 = Zonula occludens-2.

**Figure 2 cells-11-01569-f002:**
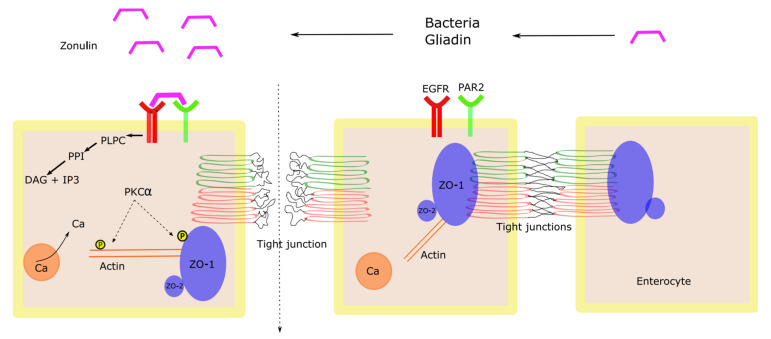
Zonulin-mediated increase in intestinal permeability. EGRF = epidermal growth factor receptor; PAR2 = proteinase-activated receptor 2; PLPC = phospholipase C; PPI = phosphatidyl inositol; DAG = diacylglycerol; IP3 = inositol 1,4,5-tris phosphate; PKCα = Protein kinase C alpha; Ca = calcium; ZO-1 = Zonula occludens-1 ZO-2 = Zonula occludens-2.

**Figure 3 cells-11-01569-f003:**
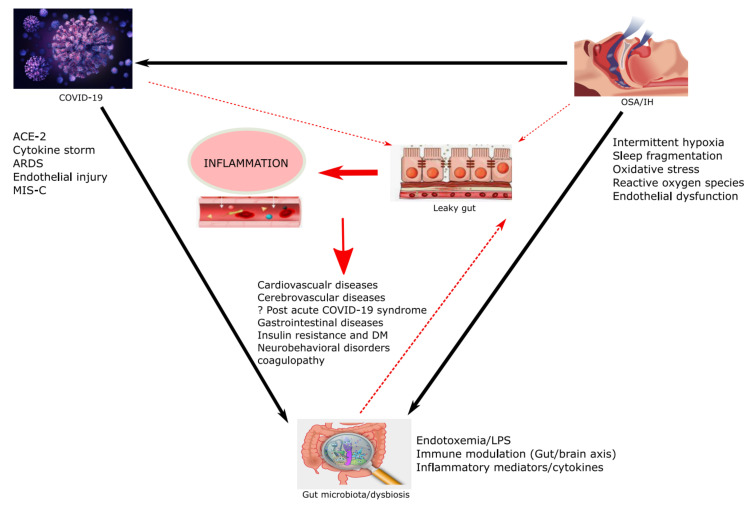
The inter-relationship between COVID-19, obstructive sleep apnea, and the gut microbiome. ACE-2 = Angiotensin-Converting Enzyme 2; MIS-C = Multisystem inflammatory syndrome in children; OSA = obstructive sleep apnea; IH = intermittent hypoxia; ARDS = acute respiratory distress syndrome; LPS = Lipopolysaccharide; DM = diabetes mellitus.

**Table 1 cells-11-01569-t001:** The impact of obstructive sleep apnea on COVID-19 outcome in hospitalized patients with COVID-19 infection.

Author, Year (Country)	Total Sample Size, OSA Sample Size, (M), (Age)	Design	BMI (mean)	Diagnostic Test (COVID-19)	Diagnostic Test (OSA)	PAP Rx	COVID-19 Outcome during Hospitalization (Statistically Significant)	Results Summary
Cariou et al., 2020 [27] (Coronado Study) France	1317, 1189, 865,70	P	28.4	PCR	Self-reported	Yes	Treated OSA was independent risk of death at day 7 after admission in COVID-19 patients ((OR 2.80 (1.46, 5.38), *p* = 0.002)	- The primary outcome was combined tracheal intubation for mechanical ventilation and/or death within 7 days of admission. - Obesity is associated with poor early prognosis in patients with T2D hospitalized for COVID-19. - This poor prognosis was not observed in elderly individuals.
Maas et al., 2020 [21] USA	9405, 592, 4316, N/A	R	N/A	PCR	EMR using ICD-10 codes	N/A	OSA was associated with a greater risk for acquiring COVID-19 infection (OR 1.65 (1.36–2.02)) and respiratory failure (OR 1.98 (1.65–2.37))	- Mean BMI was not mentioned. However, the OR of acquiring COVID-19 and respiratory failure were categorized based on BMI (overweight, class I, II, and III obesity). - ICD-10 codes used are COVID-19 (U07), OSA (G47.33), Type 2 diabetes mellitus (E11), hypertension (I10), and respiratory failure as (J96).
Strausz et al., 2021 [22] Finland	445, 38, 166, 53	R	27	PCR	- Subjective symptoms - Clinical examination - PSG (AHI ≥ 5)	Yes (EMR)	OSA was associated with higher risk of hospitalization after adjusting for sex, BMI, and co-morbidities (OR 2.93 (1.02–8.39), *p* = 0.05)	- The data of this study were obtained from a large biobank study in Finland (FinnGen study). - COVID-19 data were obtained from The National Infectious Diseases Registry in Finland. - OSA treatment data were obtained from Heart and Lung Center, or Department of Oral and Maxillofacial Diseases (Helsinki University Hospital), Finland - OSA was not associated with an increased risk for contracting COVID-19 infection.
Tak Kyu Oh et al., 2021 [26] South Korea	122040, 550, 48726, N/A	R	N/A	PCR	EMR using ICD-10 codes	N/A	OSA (with ILD) was associated with increased incidence of COVID-19 infection (OR 1.65 (1.23–2.16); *p* < 0.001)	- COPD was associated with a higher risk of hospital mortality among patients with COVID19. - BMI was not included which can be a limitation. - OSA was not associated with increased risk of hospital mortality.
Peker et al., 2021 [36] Turkey	320, N/A, 173, 53	P	27.4	PCR	Modified Berlin questionnaire.	N/A	modified high risk OSA (mHR-OSA) associated with: Clinical worsening of COVID-19 (aHR 1.55 (1.00–2.39)) Increased O_2_ requirement (OR 1.95, (1.06–3.59)) Poor outcome.	- The impact of mHR-OSA on COVID-19 was independent of age, sex, and comorbidities. - No sleep studies were conducted. Based on a modified version of Berlin questionnaire, participants were categorized into modified low or high risk for OSA.
Mashaqi et al., 2021 [31] USA	1738, 139, 851, 58	R	- 30 (Non-OSA - 40 (OSA group)	EMR using ICD-10 code	EMR using ICD-10 code	N/A	- None	- OSA was not an independent risk factor for worse COVID-19 outcome in hospitalized patients (after adjusting for demographic, BMI, and co-morbidities) - Clinical outcomes measured in hospitalized patients included (ICU admission, intubation and mechanical ventilation, mortality, and length of hospital stay).
Cade et al., 2020 [19] USA	4668, 443, 2073, 56	R	28.8	EMR using ICD-10 code	EMR using ICD-10 code	Yes (EMR)	OSA was associated with a greater risk of overall mortality.	- The results were after adjusting for demographics, BMI, and medical co-morbidities. - OSA was not associated with increased risk of intubation, ICU admission, or inpatient admission.
Kravitz et al., 2021 [23] USA	312, 312, 153, 61	R	37	PCR	EMR using ICD-10 codes	Yes (EMR)	Moderate and severe OSA was associated with higher risk of hospitalization (OR 4.29, (2.09-9.02))	- OSA severity among hospitalized patients was not associated with a composite outcome of mechanical ventilation, intensive care unit admission, and death. - Mild OSA was not associated with higher risk of hospitalization.
Goldstein et al., 2021 [30] USA	572, 113, 320, 63	R	31	PCR	PSG/HSAT	N/A	None	After adjusting for demographics and BMI, OSA was not associated with increased risk of mechanical ventilation, vasopressor requirement, length of stay, or death.
Kar et al., 2021 [37] India	213, N/A, 144,55	P	26.7	PCR	Questionnaires - STOPBANG, - Berlin - NoSAS - Epworth	N/A	Patients who died had positive screening questionnaires compared to survived patients.	Age ≥ 55 and STOPBANG score ≥ 5 were found to have small positive but independent effect on mortality even after adjusting for other variables.
Beltramo et al., 2021 [24] France	89,530, 3581, 47,495, 65	R	N/A	EMR using ICD-10 codes	EMR using ICD-10 codes	N/A	Hospitalized COVID-19 pts with OSA had higher rate of PE, VAP, ICU stay, and death compared to pts hospitalized for influenza.	The hospitalization outcome was compared between COVID-19 and Influenza patients.
Gottlieb et al., 2020 [20] USA	8673, 288, 4045, 41	R (case-control)	27.2	PCR	EMR	N/A	OSA is associated with higher risk of critical illness in COVID-19 patients.	- Age, male sex, Hispanic/Latino ethnicity, HTN, DM, prior CVA, CAD, CHF, ESRD, cirrhosis, were more commonly associated with admission. - Male sex, CHF, OSA, bloodborne cancer were associated with higher risk of critical illness.
Zhang et al., 2020 [29] China	97, N/A, 43, 58	R	24.1	PCR	EMR using symptomless multi-Variable apnea prediction (sMVAP)	N/A	↑ sMVAP on admission were associated with higher odds of in-hospital death, ICU transfer rate.	In addition to higher sMVAP, CHD, and d-dimer were associated with higher risk of hospital death.
Rögnvaldsson et al., 2021 [28] Iceland	4756, 185, 2455, 39	R	26 (non OSA) 32 (OSA) Median	PCR	Sleep database at The National University Hospital of Iceland (LUH)	Yes (49%) No (51%)	OSA was associated with twofold increase in risk of severe COVID-19	- This association between OSA and COVID-19 persisted after adjustment for several known confounding factors, including obesity. - This association between OSA and COVID-19 was not modified by PAP treatment.

R = retrospective; P = prospective ICD = international classification of diseases; EMR = electronic medical records; N/A = not available;T2D = type 2 diabetes; PCR = polymerase chain reaction; HTN = hypertension; CVA = cerebrovascular accident; CAD = coronary artery disease; CHD = coronary heart disease; CHF = congestive heart failure; ESRD = end-stage renal disease; OSA = obstructive sleep apnea; OR = odds ratio; BMI = body mass index; PSG = polysomnography; HSAT = home sleep apnea test; ILD = interstitial lung diseases; PE = pulmonary embolism; VAP = ventilator associated pneumonia; AHI = apnea-hypopnea index; aHR = adjusted hazard ratio.

**Table 2 cells-11-01569-t002:** OSA and gut dysbiosis in animal experiments.

Author, Year	OSA Model (Duration)	Lab Method	Relative Microbial Abundance	Comments
Morenos et al. 2014 [39]	IH—6 weeks	16 S r RNA	↑Prevotella ↑Paraprevotella ↑Desulfovibrio ↑Lachnospiraceae ↓Bacteroides ↓Odoribacter ↓Turicibacter ↓Peptococcaceae ↓Erysipelotrichaceae	Significant increase in Mucin-degrading bacteria.
Morenos et al. 2016 [40]	IH—6 weeks	16 S r RNA	↑Firmicutes ↑Deferribacteres ↓Bacteroidetes	IH followed by 6 weeks of normoxic recovery which did not show a significant change
Tripathi et al. 2018 [44]	IHC—6 weeks	16 S r RNA	↑Mogibacteriaceae ↑Oscillospira ↑Lachnospiraceae ↑Clostridiaceae	Intermittent hypoxia and hypercapnia instead of hypoxia alone suggesting the role of hypercapnia on dysbiosis.
Poroyko et al. 2016 [48]	SF—4 weeks	16 S r RNA	↑Firmicutes ↑Lachnospiraceae ↑Ruminococcaceae ↓Bacteroidetes ↓Actinobacteria ↓Lactobacillaceae ↓Bifidobacteriaceae	- ↑TNF-α, IL-6, LBP, Leptin and NGAL. - ↑insulin resistance. - Reversibility upon discontinuation of SF.
Liu et al. 2019 [46]	IH—6 weeks	16 S r RNA	↓*Lactobacillus rhamnosus*	Synergistic effect of IH with HFD to augment high blood pressure readings.
Durgan et al. 2016 [47]	Endotracheal implant	16 S r RNA	↓Ruminococcaceae ↓Clostridiales	- Hypertension was mainly seen in combination of OSA and HFD. - The highest F/B ratio, microbial richness and diversity were mainly seen in HFD
Ganesh et al. 2018 [49]	Endotracheal implant	16 S r RNA	↑RC4-4 ↑Akkermanisa	- Combination of OSA and HFD resulted in the highest dysbiotic changes. - Probiotic and Prebiotic helped in protecting the gut barrier.

IH = intermittent hypoxia, IHC = intermittent hypercapnia, HFD = high-fat diet, OSA = obstructive sleep apnea, F/B = Firmicutes/Bacteroidetes, LBP = lipopolysaccharide binding protein, SF = sleep fragmentation, TNF = tumor necrosis factor, IL-6 = interleukin-6, NGAL = neutrophil gelatinase-associated lipocalin.

**Table 3 cells-11-01569-t003:** COVID-19 and the gut microbiota in clinical studies.

Author, Year	Study Design	Sample Size	Lab Methods (Sequencing)	Relative Microbial Abundance	Results Summary
Gu et al. 2020 [61]	Cross-sectional	**- Total (84)** -COVID-19 (30) - H1N1 (24) - Control (30)	16S rRNA	**H1N1** ↓ *Actinobacteria* and *Firmicutes*. **COVID-19** ↑ *Streptococcus, Rothia*, *Veillonella*, *Erysipelactoclostridium*, and *Actinomyces*.	- Negative correlation with COVID-19 depleted bacteria CRP, PCT, and D-dimer level. - Positive correlation with increased bacteria, CRP, and *D*-dimer levels.
Zuo et al. 2020 [52]	Prospective	**- Total (36)** - COVID-19 (15) - CAP (6) - Control (15)	Shotgun	**Antibiotic Naïve:** Clostridium hathewayi, Actinomyces viscosus, and Bacteroides nordii. **COVID-19 (abx +):** decreased symbionts, Faecalibacterium prausnitzii, Lachnospiraceae bacterium, Bubacterium rectale, Ruminococcus obeium, and Dorea formicigenerans. **Decreased Symbionts:** Eubacterium ventriosum, Faecalibacerium prausnitzii, Roseburia, Lachnospiraceae taxa. **Increased opportunisitic pathogens**: clostridium hathewayi, Actinomyces ciscosus, Bacteroides nordii.	- ↑ *Firmicutes, Coprobacillus, Clostridium ramosum, and C. hathewayi*, in severe COVID-19. - Bacteroidetes phylum showed inverse relation to stool viral load. - Fimicutes species, Erysipelotrichaceae bacterium, showed strong positive correlation with viral load.
Zuo et al. 2020 [54]	Prospective	**- Total (69)** - COVID-19 (30) - CAP (9) - Control (30)	- Shotgun	**Increased Candida** - *C. Albicans* > 50% *- A. flavus* - *A. niger*	**COVID-19** ↑ *Candida albicans*, *C. auris, and Aspergillus flavus*. ↑ *C. albicans, C auris, and A. Flavus* also seen in CAP patients.
Zuo et al. 2020 [65]	Prospective	**- Total (15)** - COVID-19 (15)	- Shotgun	**High levels of stool SARS-CoV-2** ↑ *Collinsella aerofaciens, Collinsella tanakaei, Streptococcus infantis, Morganella moranii* **Low to absent fecal SARS-CoV-2** ↑*Parabaceroides merdae, Bacteroides stercoris, Alistipes onderdonkii and Lachnspiraceae bacterium.*	↑ opportunistic pathogens including *Collinsella aerofaciens* and *Morganella morganii* in high SARS-CoV-2 fecal samples. ↑ immune boosting bacteria including *Parabacteroides, Bacteroides*, and *Lachnospiraceae* in low to absent SARS-CoV-2 s fecal samples.
Yeoh et al. 2021 [56]	Prospective	**- Total (178)** - COVID-19 (100) - Control (78)	- Shotgun	**COVID-19** ↑ *Bacteroidetes* **COVID-19 (abx adjusted)** ↑ *Parabacteroides*, *Sutterella wadsworthensis* and *Bacteroides caccae.* ↓ *Adlercreutzia equolifaciens*, *Dorea formicigenerans* and *Clostridium leptum* **Controls** ↑ *Actinobacteria*	-*F. prausnitzii* and *Bifido- bacterium bifidum* were negatively correlated with COVID-19 disease severity when adjusted for abx use. - Known immunomodulating bacteria that are depleted in COVID-19 (such as *B. adolescentis*, *E. rectale* and *F. prausnitzii)* were inversely related to cytokine levels.*-B. dorei* and *Akkermansia muciniphila* were positively correlated with cytokines. These were enriched in COVID-19. -Gut microbiota of recovered patients were enriched in species including *Bifidobacterium dentium* and *Lactobacillus ruminis* were increased 30 days post-recovery while *E. rectale*, *R. bromii*, *F. prausnitzii* and *Bifidobacterium longum* remained depleted.
Mazzarelli et al. 2021 [57]	Prospective	**- Total (23)** - COVID-19w (9) -COVID-19i (6) -Control (8)	16S RNA	**COVID-19 vs. control** ↑ Proteobacteria phylum *Peptostreptococcaceae*, *Enterobacteriaceae*, *Staphylococcaceae*, *Vibrionaceae*, *Aerococcaceae*, *Dermabacteraceae*, *Actinobacteria* ↓ Spirochaetes and Fusobacteria phyla *Nitrospiraceae*, *Propionibacteriaceae*, *Aeromonadaceae*, *Moraxellaceae*, *Mycoplasmataceae* **COVID-19i vs. control** ↑ *Erysipelotrichaceae*, *Microbacteriaceae*, *Mycobacteriaceae*, *Pseudonocardiaceae*, *Brevibacteriaceae* ↓ Carnobacteriaceae, *Coriobacteriaceae,* and *Mycoplasmataceae* **COVID-19i vs. COVID-19w** ↑ *Staphylococcaceae*, *Microbacteriaceae*, *Micrococcaceae*, *Pseudonocardiaceae*, *Erysipelotrichales* ↓ *Carnobacteriaceae*, *Pectobacteriaceae*, *Moritellaceae*, *Selenomonadaceae*, *Micromonosporaceae*, *Coriobacteriaceae*	-ICU patients showed decreased gut microbiota diversity compared to medical floor and controls -Proinflammatory bacteria *Peptostreptococcaceae*, *Enterobacteriaceae*, *Staphylococcaceae*, *Vibrionaceae*, *Aerococcaceae*, *Dermabacteraceae*, *Actinobacteria* were increased in both ICU and wards compared to control. Ward gut microbiota closer to control. -Increase in inflammatory associated *Faecalibacterium* in ICU patients while decrease in anti-inflammatory associated *Ruminococcaceae and Clostridiaceae.* -Down regulating ACE2 bacteria, *Bacteroides dorei and Bacteroides thetaiotaomicron* were decreased in Ward patients
Zuo et al. 2021 [55]	Prospective	**- Total (176)** - COVID-19 (98) - Control (78)	- non-targeted shotgun metagenomic	**COVID-19 +** ↑ SARS-CoV-2 in fecal samples. ↓ pepper mild mottle virus (PMMoV). ↑ eukaryotic and environment prevalent viruses. ↑Streptococcus phage, Escherichia phage, Homavirus, Lactococcus phage, Ralstonia phage, Solumvirus, and Microcystis phage **COVID-19 −** ↑ bacteriophages in feces (69%, 18 out of 26 DNA virus species)	-Pepper chlorotic spot virus (PCSV) inversely correlated with COVID-19 disease severity -Fecal SARS-CoV-2 levels were significantly lower in moderate COVID-19 when compared to less severe COVID-19 -9 DNA virus species (Myxococcus phage, Rheinheimera phage, Microcystis virus, Bacteroides phage, Murmansk poxvirus, Saudi moumouvirus, Sphaerotilus phage, Tomelloso virus, and Ruegeria phage) inversely correlated with disease severity of COVID-19 and 8 showed strong negative correlation to inflammation markers in the blood (LDH, neutrophil count, white cell count, or CRP). - 4 of the 9 viral species (Myxococcus phage, Bacteroides phage, Murmansk poxvirus, and Sphaerotilus phage) were noted to inversely correlated with human age.

CRP = C-reactive protein, PCT = Prolactin, CAP = community-acquired pneumonia, Abx = antibiotic, COVID-19i = COVID-19 in ICU, COVID-19w = COVID-19 in wards, ACE2 = Angiotensin-Converting Enzyme 2.

## Data Availability

Not applicable.

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
