# Peer review of "Obstructive Sleep Apnea as a Risk Factor for COVID-19 Severity—The Gut Microbiome as a Common Player Mediating Systemic Inflammation via Gut Barrier Dysfunction"

_cells, 2022, doi:10.3390/cells11091569_

Round 1

Reviewer 1 Report

This is an interesting review summarizing the current knowledge about the relationships among gut microbiota, OSA and COVID-19. The suggestion about OSA-related therapeutic options for COVID-19 is sound and deserved further consideration by the scientific community.

In my opinion, the manuscript deserves publication, but some improvements could be achieved:

  1. the text is very long, and I think it could be shortened. The description of the results from the many cited studies could be synthesized and probably limited to the most important ones, referring to the Tables for the complete lists of the studies.
  2. discussion, page 22, lines 221-239. Targeting the gut microbiome does not seem actionable since no definite hallmarks for targeted intervention are available. This part should be re-write focusing on current limits of microbiome studies and microbiome shaping but highlighting the potential to uncover novel research directions.
  3. conclusion, page 22, lines 205-207. Please mitigate this statement.

Minor comments:

- Table 1. I suggest including OSA sample size in column 2 for easier reading.

- some typos are present, Check page 2, line 70; page 3 line 103 (add ICU definition), line 121, page 4 line 154, 173

- page 10, lines 56. I think results from reference 47-50 have not been exposed.

-page 18 line 58. In my opinion, you should state that ref 79 is a preprint paper

Author Response

Reviewer 1

 Dear reviewer:

  We would like to extend our gratitude for the great feedback and the valuable comments you provided. We believe these comments will further enrich the review. Please see below our response to each comment in red

Comments and Suggestions for Authors

This is an interesting review summarizing the current knowledge about the relationships among gut microbiota, OSA and COVID-19. The suggestion about OSA-related therapeutic options for COVID-19 is sound and deserved further consideration by the scientific community.

In my opinion, the manuscript deserves publication, but some improvements could be achieved:

  1. the text is very long, and I think it could be shortened. The description of the results from the many cited studies could be synthesized and probably limited to the most important ones, referring to the Tables for the complete lists of the studies.

  Thank you for the comment. We agree. We made a significant change in section 2 (OSA and COVID-19 (Is there a plausible link clinically?). We shortened the text and relied more on table 1 especially that table 1 is detailed. In sections 3 and 4, the cut in the text was less since we felt that some degree of details is still needed in relation to COVID-19 and the gut microbiome to make sure that the reader will the get point from each study.

  1. discussion, page 22, lines 221-239. Targeting the gut microbiome does not seem actionable since no definite hallmarks for targeted intervention are available. This part should be re-write focusing on current limits of microbiome studies and microbiome shaping but highlighting the potential to uncover novel research directions.

  Thank you for the comment. We added a sentence at the beginning of this paragraph to reflect those therapeutic interventions targeting the gut microbiome are still novel and not implemented clinically.

  1. conclusion, page 22, lines 205-207. Please mitigate this statement.

  Thank you for the comment. We concised the conclusion section.

Minor comments:

- Table 1. I suggest including OSA sample size in column 2 for easier reading.

    Thank you for the comment. We added the OSA sample size to column 2.

- some typos are present, Check page 2, line 70; page 3 line 103 (add ICU definition), line 121, page 4 line 154, 173

    Thank you. Typos corrected.

- page 10, lines 56. I think results from reference 47-50 have not been exposed.

    Thank you for the comment. We removed these references from the manuscript.

-page 18 line 58. In my opinion, you should state that ref 79 is a preprint paper

    Appreciate this valuable comment. We added a note indicating that this reference is a preprint.

Reviewer 2 Report

This is a critical review about the role of Gut Microbiome in Obstructive Sleep Apnea among patients with COVID-19. The topic is very interesting and of clinical significance. There were some suggestions:

  1. Suggest to check English grammar carefully. For example, line 23 ” Several risk factors have been shown to worsen the severity…..” should be ” Several risk factors have been shown to worse the severity…..”
  2. Line 24-26 “Obstructive sleep apnea …..shares many of these risk factors raising the possibility that OSA is an independent risk factor for COVID-19 severity” This sentence is difficult to understand. Please clarify this sentence.
  3. In Introduction part, suggest focus on why link microbiota to COVID19 and OSA? Especially in line 82 “we explore the hypothesis that the gut……..”. Why have this hypothesis?
  4. Line 100-146. As a review article, suggest to analyze these studies and make some conclusions or suggestions instead of just listing these studies.
  5. Suggest to uniform the location of citation at the end of the sentence.
  6. Line 147-167 “In contradistinction, some studies…….” As a review article, suggest discuss why these were discrepancy over these articles.
  7. As for “OSA and gut dysbiosis (Is there evidence?)” (line 1-67) All of these descriptions were related to oxygen but not OSA. Please revise this part.
  8. For “OSA and SARS-CoV-2, similar pathological features on the gut barrier at the cellular and molecular level” part. The connection and mechanism are too arbitrary. Suggest revised this part.
  9. Overall though the topic is very interesting, the evidence mentioned in this article is too weak to link the connections between microbiota, COVID19 and OSA.

Author Response

Reviewer 2

We would like to extend our gratitude for the great feedback and the valuable comments you provided. We believe these comments will further enrich the review. Please see below our response to each comment in red

Comments and Suggestions for Authors

This is a critical review about the role of Gut Microbiome in Obstructive Sleep Apnea among patients with COVID-19. The topic is very interesting and of clinical significance. There were some suggestions:

  1. Suggest checking English grammar carefully. For example, line 23 ” Several risk factors have been shown to worsen the severity…..” should be ” Several risk factors have been shown to worse the severity…..”

  Thank you for the comment. We reviewed the paper another time for any grammar mistakes. We would like to share with you the example you listed (line 23) since we believe the statement, we included is grammatically correct “Several risk factors have been shown to worsen the severity…...” Please correct us if otherwise.  

  1. Line 24-26 “Obstructive sleep apnea …..shares many of these risk factors raising the possibility that OSA is an independent risk factor for COVID-19 severity” This sentence is difficult to understand. Please clarify this sentence.

  Thank you for the comment. We re-phrased the sentence.

  1. In Introduction part, suggest focus on why link microbiota to COVID19 and OSA? Especially in line 82 “we explore the hypothesis that the gut……..”. Why have this hypothesis?

  Thank you for this comment. We discussed this in detail in the discussion section, but we totally agree that mentioning it in the introduction is engaging to the reader. We added a statement after this statement “we explore the hypothesis that the gut……..”.

  1. Line 100-146. As a review article, suggest to analyze these studies and make some conclusions or suggestions instead of just listing these studies.

  Thank you for the comment. We cut the whole section and concised it in a more comprehensive way and relied more on table 1 since all these studies are listed in detail in the table.

  1. Suggest to uniform the location of citation at the end of the sentence.

 Edited citations as suggested. Thanks

  1. Line 147-167 “In contradistinction, some studies…….” As a review article, suggest discuss why these were discrepancy over these articles.

   Thank you for the comment. Added a short paragraph to discuss this discrepancy.

  1. As for “OSA and gut dysbiosis (Is there evidence?)” (line 1-67) All of these descriptions were related to oxygen but not OSA. Please revise this part.

  Thank you very much. All these experiments in mice or rats used intermittent hypoxia which is the model used to mimic OSA. They exposed these animals to oxygen at different (PaO2) in a hypoxia chamber for a specific duration and at specific frequency. Few experiments used an implantable device in the neck that is inflated/deflated to mimic OSA.

  1. For “OSA and SARS-CoV-2, similar pathological features on the gut barrier at the cellular and molecular level” part. The connection and mechanism are too arbitrary. Suggest revised this part.

  Thank you for the comment. Revised the whole section. Focused on the biomarkers of gut barrier dysfunction in OSA and COVID-19 as a common feature and added two more references at the end.

  1. Overall though the topic is very interesting, the evidence mentioned in this article is too weak to link the connections between microbiota, COVID19 and OSA.

   Thank you for the comment. We totally respect this opinion. We admit the limitations to our review, the main one is that it is narrative review. However, we tried to include all the literature written in relation to this triangular inter-relationship and most of the studies included favor this relationship. We did not conduct a systematic review or meta-analysis (which provides a higher level of evidence) but we believe our review is the first one written to shed the light on such conceptual relationship between OSA, COVID-19 and the gut microbiome. We have to admit also that most of the COVID-19 studies were retrospective in nature which limited the strength of evidence in this regard.

Round 2

Reviewer 2 Report

The article had been revised as suggestions.